# Development and validation of systems for genetic manipulation of the Old World tick-borne relapsing fever spirochete, *Borrelia duttonii*

Clay D. Jackson-Litteken[1�¤], Wanfeng Guo[1�], Brandon A. Hogland[1], C. Tyler Ratliff[1], LeAnn McFadden[2], Marissa S. Fullerton[1], Daniel E. Voth[1], Ryan O. M. Rego[3,4], Jon S. Blevins[1]*

**1** Department of Microbiology and Immunology, University of Arkansas for Medical Sciences, Little Rock, Arkansas, United States of America, **2** Department of Biology, University of Arkansas at Little Rock, Little Rock, Arkansas, United States of America, **3** Institute of Parasitology, Biology Centre CAS, České Budějovice, Czech Republic, **4** Faculty of Science, University of South Bohemia, České Budějovice, Czech Republic

উ These authors contributed equally to this work.
¤ Current address: Department of Molecular Microbiology, Washington University School of Medicine, St. Louis, Missouri, United States of America
* jsblevins@uams.edu

**Data Availability Statement:** All data are in the manuscript and/or supporting information files.

## Abstract

Relapsing fever (RF), a vector-borne disease caused by *Borrelia* spp., is characterized by recurring febrile episodes due to repeated bouts of bacteremia. RF spirochetes can be geographically and phylogenetically divided into two distinct groups; Old World RF *Borrelia* (found in Africa, Asia, and Europe) and New World RF *Borrelia* (found in the Americas). While RF is a rarely reported disease in the Americas, RF is prevalent in endemic parts of Africa. Despite phylogenetic differences between Old World and New World RF *Borrelia* and higher incidence of disease associated with Old World RF spirochete infection, genetic manipulation has only been described in New World RF bacteria. Herein, we report the generation of genetic tools for use in the Old World RF spirochete, *Borrelia duttonii*. We describe methods for transformation and establish shuttle vector- and integration-based approaches for genetic complementation, creating green fluorescent protein (*gfp*)-expressing *B. duttonii* strains as a proof of principle. Allelic exchange mutagenesis was also used to inactivate a homolog of the *Borrelia burgdorferi p66* gene, which encodes an important virulence factor, in *B. duttonii* and demonstrate that this mutant was attenuated in a murine model of RF. Finally, the *B. duttonii p66* mutant was complemented using shuttle vector- and *cis* integration-based approaches. As expected, complemented *p66* mutant strains were fully infectious, confirming that P66 is required for optimal mammalian infection. The genetic tools and techniques reported herein represent an important advancement in the study of RF *Borrelia* that allows for future characterization of virulence determinants and colonization factors important for the enzootic cycle of Old World RF spirochetes.

**Funding:** This work was supported by grants from the National Institutes of Health/National Institute of Allergy and Infectious Diseases (R03AI151432; JSB), the National Institute of General Medical Sciences (P20GM103625 and P30GM145393; JSB), the UAMS Vice Chancellor of Research & Innovation (JSB), and the Arkansas Biosciences Institute (JSB). The funders had no role in study design, data collection and analysis, decision to publish, or preparation of the manuscript.

**Competing interests:** The authors have declared that no competing interests exist.

## Author summary

Relapsing fever is a globally distributed, vector-borne bacterial infection that is characterized by recurring febrile episodes. The causative *Borrelia* spp. were identified over 100 years ago, but little is known regarding factors required for mammalian infection or vector colonization/transmission. Relapsing fever *Borrelia* can be geographically and phylogenetically divided into distinct clades; New World, found in the Americas, and Old World, found in Africa, Europe, and Asia. Although Old World relapsing fever *Borrelia* represent major causes of disease in endemic regions of the world, genetic studies aimed at identifying bacterial gene products required during the tick-vertebrate infectious cycle have focused on the less commonly reported New World relapsing fever *Borrelia*. Herein, we begin to address this knowledge gap by developing techniques and molecular tools for genetic manipulation of the Old World tick-borne relapsing fever spirochete, *Borrelia duttonii*. This genetic system will lay the foundation for future studies aimed at identifying bacterial factors required during the enzootic cycle of *B. duttonii*.

## Introduction

*Borrelia* spp. are globally distributed, vector-borne bacteria that cause diverse diseases, rendering these pathogens an immense public health concern [1,2]. *Borrelia* can be divided into two distinct groups, relapsing fever (RF) and Lyme disease (LD) *Borrelia*, based on associated disease courses, vector/enzootic cycles, and additional phylogenetic differences [1–3]. Infection with RF spirochetes is characterized by recurring severe, febrile episodes that coincide with repeated bouts of high-level bacteremia (up to $10^8$ bacteria/ml of blood) [1,4–8]. Furthermore, infection by RF *Borrelia* can lead to more severe symptoms and outcomes, such as acute respiratory distress syndrome, jaundice, meningitis, splenomegaly, and perinatal mortality [4,9,10]. Alternatively, mammalian infection with LD *Borrelia* results in lower levels of bacteremia ($10^3$–$10^4$ bacteria/ml of blood) that is restricted to early stages of infection, after which it disseminates into multiple host tissues [11]. Therefore, symptoms of RF are primarily associated with acute bacteremia, whereas most LD symptoms occur because of chronic tissue colonization (e.g., arthritis, neuroborreliosis, and carditis) [2]. In addition, arthropod vectors differ greatly between RF and LD *Borrelia*. RF bacteria are transmitted by the human body louse [louse-borne relapsing fever (LBRF)] or *Ornithodoros* or *Ixodes* spp. ticks [tick-borne relapsing fever (TBRF)], while LD spirochetes are only transmitted by *Ixodes* spp. ticks [12–17]. The different disease courses resulting from infection by LD and RF *Borrelia* and their disparate enzootic cycles involving distinct vectors suggest that these two groups of bacteria have likely evolved to encode different factors required for vector acquisition and colonization, transmission, and vertebrate infection. Because most molecular research has focused primarily on LD *Borrelia*, a significant gap exists in our understanding of the biology of RF spirochetes [18,19].

RF *Borrelia* can be further subdivided into New World (found in the Americas) and Old World (found in Africa, Asia, and Europe) species [4,5,16,20]. New World RF spirochetes include *Borrelia turicatae*, which is found in the arid climate of the southwestern United States, and *Borrelia hermsii* and *Borrelia parkeri*, the causes of RF in the mountainous regions of the western United States and southwestern Canada [4,5,12,13,21–23]. Old World RF spirochetes include *Borrelia crocidurae* and *Borrelia duttonii*, which are endemic to western and eastern Africa, respectively, *Borrelia persica*, a causative agent of TBRF in Eurasia, and *Borrelia recurrentis*, the LBRF spirochete responsible for epidemic RF in eastern Africa [24–30]. RF spirochetes were historically classified into New World and Old World species based solely on

geographic separation, but phylogenetic analyses have revealed genomic differences in these bacteria, supporting the separation of the groups into two distinct clades [26,31–34].

Although RF is globally distributed, the public health impact varies throughout the world. While RF is a relatively rare and possibly underreported disease in the Americas, this infection is endemic and highly prevalent in parts of Africa [1,4,5,20,25–27,32]. In fact, RF has been reported as the most common bacterial infection in Senegal, the most prevalent cause of fever in rural Zaire, and a top 10 cause of death in children under the age of 5 in Tanzania [1,25,27,35–37]. Furthermore, cases are often unreported or are misdiagnosed as another disease, such as malaria, in endemic regions of Africa, implying that the actual incidence is higher than previously reported [28,38]. Despite the clear public health threat that Old World RF spirochetes represent, very little is known regarding the molecular pathogenesis of these bacteria.

Use of molecular Koch's postulates provides the most direct way to determine a causal relationship between a bacterial gene, the protein it encodes, and a requirement during the bacterial lifecycle [39,40]. To apply molecular Koch's postulates, the gene of interest is inactivated, and the phenotype of the mutant is compared to that of the wild-type parent. Additionally, expression of the gene must be restored in the mutant to assess if the phenotype of interest is reversed. While a limited number of studies have applied genetic manipulation in RF *Borrelia* to investigate proteins potentially required during the bacteria's enzootic cycle, these studies have only been conducted with the New World RF spirochetes, *B. hermsii* and *B. turicatae* [41–50]. Extensive geographic separation and phylogenetic differences between New World and Old World RF spirochetes suggests that these two groups likely evolved distinct regulatory pathways and proteins required for mammalian virulence, as well as vector acquisition, colonization, and transmission. Therefore, there is a significant need to develop molecular techniques for the study of Old World RF *Borrelia*.

Herein, we aimed to develop a genetic system for use in the Old World TBRF spirochete, *B. duttonii*. First, we developed shuttle vector- and integration-based constructs and methods for electroporation with *B. duttonii*, creating green fluorescent protein (*gfp*)-expressing *B. duttonii* strains to validate the approach. Allelic exchange mutagenesis was then used to inactivate the gene encoding the *B. duttonii* homolog of P66, a known virulence factor of the LD spirochete, *Borrelia burgdorferi* [51,52]. Using a murine model of RF, we demonstrated that the *B. duttonii* *p66* mutant was attenuated. Finally, this *p66* mutant was complemented using shuttle vector- and *cis* integration-based approaches to restore production of P66 and murine infectivity of the *p66* mutant strains. The techniques described in this manuscript represent a foundation for the application of molecular tools to study Old World RF *Borrelia* and will ultimately guide research to better understand these understudied pathogens.

## Methods

### Ethics statement

Murine infections and immunization protocols were performed in accordance with the Guide for the Care and Use of Laboratory Animals, the Public Health Science Policy on Humane Care and Use of Laboratory Animals, and the Animal Welfare Act, and the protocol used was approved by the University of Arkansas for Medical Sciences (UAMS) Institutional Animal Care and Use Committee (IACUC). UAMS is accredited by the International Association for Assessment and Accreditation of Laboratory Animals Care (AAALAC).

### Bacterial strains and culture conditions

Bacterial strains used in this study are listed in Table 1. *Escherichia coli* strain Top10F' (Life Technologies, Carlsbad, CA) was used for cloning and plasmid propagation and strain

**Table 1. Plasmids and strains used in this study.**

| Plasmid or Strain | Description[a] | Source |
|---|---|---|
| Plasmid | | |
| pGEM-T Easy | TA cloning vector; Amp[r] | Promega |
| pProEX-HTb | Expression plasmid; N-terminal His$_6$ tag; Amp[r] | Invitrogen |
| pUAMS4 | pGEM-T Easy::P*flgB*-*aacC1* (For *B. turicatae*) (AscI-flanked); Gent[r], Amp[r] | [46] |
| pUAMS353 | pGEM-T Easy::ColE1 ori (AscI/BglII-flanked); Amp[r] | This study |
| pUAMS356 | pGEM-T Easy::P*flgB*-*aacC1* (For *B. duttonii*) (AscI-flanked); Gent[r], Amp[r] | This study |
| pUAMS359 | pGEM-T Easy::P*flaB*-*aphI* (For *B. duttonii*) (BglII/AscI-flanked); Kan[r], Amp[r] | This study |
| pJD44 | *aph*[3′]-*IIIa* marked derivative of pBSV2; Kan[r] | [55] |
| pUAMS363 | pJD44::P*flaB*-*aphI* (From pUAMS359); Kan[r] | This study |
| pUAMS364 | pGEM-T Easy::pl41 ori (BamHI/AscI-flanked); Amp[r] | This study |
| pBdSV | *B. duttonii* shuttle vector; Kan[r] | This study |
| pBdSV[CCW] | pBdSV with *aphI* marker and MCS in the reverse orientation; Kan[r] | This study |
| pGreenTIR | Prokaryotic *gfp* reporter with F64L/S65T mutations and modified translation initiation region; Amp[r] | [56, 57] |
| pUAMS369 | pGEM-T Easy::::P*flaB*-*gfp* (BamHI/HindIII-flanked); Amp[r] | This study |
| pBdSV::*gfp* | pBdSV containing the P*flaB*-*gfp* cassette; Kan[r] | This study |
| pUAMS376 | pGEM-T Easy with 5' and 3' flanking regions for pl165 integration; Amp[r] | This study |
| pBd*gfp* | pUAMS376::*gfp*-*aphI* cassette; Kan[r], Amp[r] | This study |
| pBdΔ*p66* | *p66* mutational construct; Gent[r], Amp[r] | This study |
| pBdSV::*p66* | pBdSV containing *p66* with the putative promoter region; Kan[r] | This study |
| pUAMS521 | pGEM-T Easy::P*flaB*-*aadA* (For *B. duttonii*) (AscI-flanked); Amp[r], Spec[r] | This study |
| pBd*p66*$^{Cis}$ | pGEM-T Easy::*p66 cis* complementation construct with *aadA*; Amp[r], Spec[r] | This study |
| pUAMS481 | pProEX-HTb::*p66*$^{(21-612)}$; Amp[r] | This study |
| Strain | | |
| *E. coli* | | |
| TOP10F' | F′ [*lacI*$^q$Tn*10*(Tet[r])] *mcrA* Δ(*mrr-hsdRMS-mcrBC*) φ80*lacZ*ΔM15 *nupG* Δ*lacX74 recA1 ara*Δ*139* Δ(*ara-leu*)*7697 galU galK rpsL* (Strep[r]) *endA1* | Life Technologies |
| LOBSTR-BL21(DE3)-RIL | BL21(DE3) with mutated *arnA* and *slyD* and extra copies of the *argU*, *ileY*, and *leuW* tRNA genes; Cam[r] | Kerafast, Inc. |
| *B. duttonii* | | |
| BdWT | *B. duttonii* strain 1120K3, tick isolate | [31] |
| BdSV | BdWT harboring pBdSV; Kan[r] | This study |
| BdSV[CCW] | BdWT harboring pBdSV[CCW]; Kan[r] | This study |
| BdSV::*gfp* | BdWT harboring pBdSV::*gfp*; Kan[r] | This study |
| Bd*gfp* | BdWT with *gfp*-*aphI* alleles inserted into pl165; Kan[r] | This study |
| BdΔ*p66* | BdWT *p66* mutant; Gent[r] | This study |
| BdΔ*p66*$^{SV\ comp}$ | BdΔ*p66* complemented with pBdSV::*p66*; Gent[r], Kan[r] | This study |
| BdΔ*p66*$^{Cis\ comp}$ | BdΔ*p66* complemented with pBd*p66*$^{Cis}$; Strep[r] | This study |

[a]Amp, ampicillin; Gent, gentamicin; Kan, kanamycin; Spec, spectinomycin, Strep, streptomycin; Cam, chloramphenicol

LOBSTR-BL21 (DE3)-RIL (Kerafast, Boston, MA) was used for recombinant protein expression. *E. coli* was grown in lysogeny broth (LB) medium supplemented with 100 μg/ml ampicillin, 5 μg/ml gentamicin, 100 μg/ml spectinomycin, 50 μg/ml kanamycin, or 30 μg/ml chloramphenicol where appropriate. The infectious *Ornithodoros moubata* tick isolate, *B. duttonii* strain 1120K3 (designated BdWT), was used in this study [53]. *B. duttonii* was cultured in liquid modified Barbour-Stoenner-Kelly (mBSK) medium supplemented with 6 or 12% rabbit

serum at a pH of 7.6 at 35°C and 3% $CO_2$ [41,54]. *B. duttonii* was grown in the presence of 40 μg/ml gentamicin, 100 μg/ml streptomycin, or 150 μg/ml kanamycin when necessary.

## Generation of *B. duttonii* shuttle vectors

Primers used in this study are listed in S1 Table. Plasmid and genomic DNA (gDNA) preparations were performed using the Wizard Plus SV Miniprep DNA Purification System (Promega Corp, Fitchburg, WI), and PCR amplifications for cloning were conducted using high-fidelity PrimeSTAR Max DNA Polymerase (TaKaRa Bio, Mountain View, CA). All plasmids and cloned fragments were Sanger sequenced to confirm no mutations were introduced during cloning. Primers and gene nomenclature are based on the *B. duttonii* strain Ly sequence from Lescot et al. available on NCBI (GenBank assembly accession: GCA_000019685.1) [58]. Multiple-sequence alignments of *B. hermsii* and *B. duttonii* plasmid sequences were generated using MUSCLE in MacVector 18.6.4 (MacVector Inc., Apex, North Carolina).

To make the P*flaB-aphI* resistance marker for the *B. duttonii* shuttle vector, the putative promoter region for the *flaB* gene (*bdu_150*) was amplified from BdWT gDNA (primers: 5' BdP*flaB* and 3' BdP*flaB*-SOE), and the *aphI* open reading frame (ORF) was amplified from pBSV2 (primers: 5' *aphI*-SOE/Bd and 3' UAMS-88 Kan AscI) [59]. The promoter and ORF amplicons were then fused by overlap extension PCR, generating the final P*flaB-aphI* resistance marker flanked by 5' BglII and 3' AscI restriction sites. The P*flaB-aphI* marker was then TA-cloned into pGEM-T Easy (Promega), and the resulting plasmid intermediate was designated pUAMS359. The P*flaB-aphI* resistance marker from pUAMS359 was subsequently ligated into the *B. burgdorferi* shuttle vector, pJD44, replacing the *aph*[3']-*IIIa* gene, and generating the plasmid pUAMS363 [55,59,60]. The *E. coli* ColE1 origin of replication (ori) was amplified from pJD44, (primers: 5' pJD44 ColE1 ori and 3' pJD44 ColE1 ori) flanked by 5' AscI and 3' BglII restriction sites and TA-cloned into pGEM-T Easy, designated pUAMS353. The putative ori of the *B. duttonii* pl41 plasmid was amplified from BdWT gDNA (primers: 5' Bd pl41 ori BamHI and 3' Bd pl41 ori AscI) flanked by 5' BamHI and 3' AscI restriction sites and TA-cloned into pGEM-T Easy, designated pUAMS364. To generate the final *B. duttonii* shuttle vector, designated pBdSV, the AscI-flanked multiple cloning site (MCS) and P*flaB-aphI* cassette from pUAMS363, the AscI/BglII-flanked ColE1 ori from pUAMS353, and the BamHI/AscI-flanked putative *B. duttonii* pl41 ori from pUAMS364 were ligated together. Another version of the shuttle vector was additionally made in which the MCS and *aphI* resistance cassette from pUAMS363 were ligated into the shuttle vector in the opposite orientation of that in pBdSV, designated pBdSV^CCW.

To make a *B. duttonii*-adapted *gfp* cassette, the *flaB* promoter was amplified from BdWT gDNA (primers: 5' Bd*flaB*/GFP and 3' Bd*flaB*/GFP_SOE) and the *gfp* ORF was amplified from pGreenTIR (primers: 5' GFPtir ORF and 3' GFPtir ORF) [56, 57]. The *flaB* promoter and *gfp* ORF were then fused by overlap extension PCR, and the resulting product was TA-cloned into pGEM-T Easy, designated pUAMS369. The P*flaB-gfp* cassette was then excised from pUAMS369 with BamHI and HindIII and ligated into the MCS in pBdSV, generating the shuttle vector pBdSV::*gfp*.

## Generation of the pl165-specific integration construct

To create the pl165 integration construct, 5' (primers: 5' F1 pl165 int_v2 and 3' F1 pl165 int_v2.2) and 3' (primers: 5' F2 pl165 int_v2.2 and 3' F2 pl165 int_v2) regions flanking the integration site (907- and 1096-bp in size, respectively) were first amplified and TA-cloned into pGEM-T Easy. The flanking regions were then ligated together with an AscI restriction site between them in pGEM-T Easy, generating the plasmid pUAMS376. To create the *gfp*

integration construct, pBd*gfp*, a segment of pBdSV::*gfp* containing the P*flaB-gfp* and P*flaB-aphI* cassettes was excised with AscI and ligated into the AscI site in pUAMS376.

## Generation of *B. duttonii p66* mutational and complementation constructs

To generate the P*flgB-aacC1* gentamicin-resistance marker, the putative promoter region for the *flgB* gene (*bdu_297*) was amplified from BdWT gDNA (primers: 5' BdP*flgB*-AscI and 3' BdP*flgB*-SOE), and the *aacC1* ORF was amplified from pUAMS4 (primers: 5' Gent-SOE/Bd and 3' Gent-AscI/Bd) [46,47]. The promoter and marker amplicons were fused by overlap extension PCR, generating the P*flgB-aacC1* resistance marker flanked by AscI restriction sites. The P*flgB-aacC1* marker was subsequently ligated into pGEM-T Easy and designated pUAMS356.

The allelic exchange mutagenesis construct to inactivate the *p66* homolog of *B. duttonii* (e.g., *bdu_604*) was made by first amplifying upstream (primers: 5' F1 Bd P66 and 3' F1 Bd P66_AscI; size: 1098 bp) and downstream (primers: 5' F2 Bd P66_AscI and 3' F2 P66_BssHII; size: 1032 bp) regions flanking the mutation site and TA-cloning the respective amplicons into pGEM-T Easy. Flanking regions were then ligated together with the P*flgB-aacC1* resistance marker from pUAMS356 between them using the AscI restriction site to yield the final *p66* mutational construct, designated pBdΔ*p66*.

To create the BdΔ*p66* shuttle vector complementation construct, the *p66* ORF and 540-bp upstream of the coding region were amplified from BdWT gDNA (primers: 5' Bd P66 comp BamHI and 3' Bd P66 comp BamHI), and the resulting amplicon was ligated into the pBdSV MCS at its unique BamHI site. The resulting clone was designated pBdSV::*p66*. To make the *p66 cis* integration complementation construct, an *aadA* resistance marker was first adapted for use in BdWT. To generate the P*flaB-aadA* streptomycin-resistance marker, a 284-bp region upstream of the flagellin (*flaB*) gene (*bdu_150*) was amplified from BdWT gDNA (primers: 5' BdP*flaB*-AscI_long and 3' BdP*flaB-aadA* SOE), and the *aadA* ORF was amplified from a derivative of the *B. burgdorferi flgBp-aadA-trpLt* streptomycin/spectinomycin resistant cassette described by Jackson-Litteken et al. and Drecktrah et al. (primers: 5' BdP*flaB-aadA* SOE and 3' *aadA* ORF term-AscI) [45,46,61]. The resulting amplicons were fused by overlap extension PCR to generate a P*flaB-aadA* resistance marker flanked by AscI restriction sites; this P*flaB-aadA* marker was then ligated into pGEM-T Easy and designated pUAMS521. The genomic regions corresponding to a portion of *bdu_602* and all of *bdu_603* (primers: 5' F1 Bd P66 and 3' F1 Bd P66 *cis* comp; size: 990 bp) or all of *bdu_604*/*p66* and a portion of *bdu_605* (primers: 5' F2 Bd P66 *cis* comp and 3' F2 Bd P66_BssHII; size: 2894 bp) were then amplified and cloned into pGEM-T Easy. The two fragments were ligated together in pGEM-T Easy to introduce an AscI site in the *bdu_603* and *bdu_604* intergenic region, and the P*flaB-aadA* marker from pUAMS521 was ligated into this unique AscI site. The resulting *p66 cis* integration complementation construct was designated pBd*p66*$^{Cis}$.

## Transformation of *B. duttonii* and confirmation of transformants

Competent cell preparation and electroporation of *B. duttonii* were performed via previously described methods for *B. hermsii* and *B. turicatae* [41,47]. Briefly, the constructs of interest were electroporated into the appropriate strain. Following a 24-hr recovery period in mBSK medium without selection, transformants were selected using corresponding antibiotic treatment. Once viable spirochetes were observed in the post-transformation recovery cultures, individual clones were then isolated by serial-dilution plating. Transformants were confirmed by PCR, and products were separated by electrophoresis in a 0.8% agarose gel and visualized by ethidium bromide staining. GeneRuler DNA Ladder Mix (ThermoFisher Scientific, Waltham, MA) served as the molecular weight standard.

Clones transformed with shuttle vector (e.g., BdSV, BdSV[CCW], and BdSV::*gfp*) were initially confirmed by plasmid recovery in *E. coli*, restriction digest, and Sanger sequencing. Additionally, PCR was performed with gDNA from these clones to amplify internal regions of *aphI* (primers: 5' *aphI* diag and 3' *aphI* diag), *gfp* (primers: 5' GFPtir diag and 3' GFPtir diag), and *flaB* (primers: 5' BdFlaB and 3' BdFlaB). PCR was also performed to amplify an internal region of pl41 to confirm that the endogenous plasmid was not lost (primers: 5' pl41 diag and 3' pl41 diag) in shuttle vector transformants.

Successful integration of *gfp* into pl165 (e.g., Bd*gfp*) was confirmed by PCR to amplify internal regions of *gfp*, *aphI*, and *flaB* (see above). PCR was also performed to link a region upstream of the insertion site to the *gfp* allele (primers: 5' pl165 IG diag and 3' GFPtir diag) and a region downstream of the insertion site to the *aphI* allele (primers: 5' *aphI* diag and 3' pl165 IG diag).

Confirmation of the BdΔ*p66* mutant was performed by PCR to amplify an internal segment of *p66* (primers: 5' Bd P66 diag and 3' Bd P66 diag), an internal region of *aacC1* (primers: 5' *aacC1* diag and 3' *aacC1* diag), and an internal region of *flaB* (see above). To confirm the pBdSV::*p66*-transformed BdΔ*p66* (e.g., BdΔ*p66*[SV comp]), clones were initially confirmed by plasmid recovery in *E. coli*, restriction digest, and Sanger sequencing. Additionally, PCR with gDNA from shuttle vector-complemented clones was performed to amplify internal regions of *p66*, *aacC1*, and *flaB* (see above). To confirm BdΔ*p66* clones transformed with pBd*p66*[Cis] (e.g., BdΔ*p66*[Cis comp]), PCR with gDNA from transformants was performed to amplify internal regions of *p66*, *aacC1*, and *flaB* (see above), as well as *aadA* (primers: 5' *aadA* diag and 5' *aadA* diag).

## Imaging of *B. duttonii*

Imaging of *gfp*-expressing bacteria was performed as previously described [45]. Briefly, strains of interest were grown to late-exponential phase, centrifuged at ~9,400 x g for 5 min, and resuspended in PBS-MgCl$_2$. This step was repeated one more time, after which bacteria were spotted on a 1% agarose pad and covered with a coverslip [62]. Brightfield and fluorescent images were then captured using an Eclipse Ti2 microscope (Nikon Corporation, Tokyo, Japan) under 60X magnification with oil immersion.

## Production of recombinant *B. duttonii* P66 protein

The coding region for the *B. duttonii p66* homolog (BDU_604) was amplified using primers that introduced BamHI and SpeI restriction sites on the 5' and 3' ends, respectively (primers: 5' Bd P66 ORF BamHI and 3' Bd P66 ORF SpeI). The primers amplified a region of *B. duttonii* P66 corresponding to amino acids 21–612 of the predicted coding region, which was then ligated into pProEX-HTb (Life Technologies) to generate pUAMS481. To produce recombinant P66 protein, pUAMS481 was transformed in *E. coli* LOBSTR-BL21(DE3)-RIL. Expression and purification of recombinant P66 were performed as previously described, except all buffers were adjusted to a final pH 9.0 [45]. Final elution samples were buffer exchanged to 1X phosphate buffered saline (PBS), 2M urea, 5% SDS and concentrated with a 50-kDa molecular weight cutoff (MWCO) Amicon Ultra Centrifugal Filter (MilliporeSigma, Burlington, MA). The concentration of purified recombinant P66 was determined with a DC protein assay kit (Bio-Rad Laboratories, Hercules, CA).

## Generation of P66-specific antisera

The antigen-adjuvant emulsion was generated by combining 25 µg of recombinant P66 in 200 µL of sterile 1X PBS with an equal volume of AddaVax adjuvant (InvivoGen, San Diego,

CA). Three- to four-week-old, female Sprague-Dawley rats (Envigo, Indianapolis, IN) were injected subcutaneously at two sites with 200 μL of emulsion per injection. Rats were boosted twice at four-week intervals with 25 μg of recombinant protein and AddaVax as described above. Rats were euthanized and serum was collected two weeks after the second boost [45].

## Murine infections

The murine model of *B. duttonii* infection was adapted from a previous model used for *B. turicatae* and *B. hermsii* [45,46,63–65]. BdWT, BdΔ*p66*, BdΔ*p66*$^{SV\ comp}$, and BdΔ*p66*$^{Cis\ comp}$ were grown to mid-log phase, enumerated by dark-field microscopy, and diluted in fresh mBSK media to a concentration of $10^3$ spirochetes/mL. Four- to six-week-old Swiss Webster mice (Charles River Laboratories, Wilmington, MA) were injected intradermally/subcutaneously in the thoracic region with 100 μL of suspension containing a total of $10^2$ bacteria. On days 3–14 post-infection, 2.5 μL blood samples were collected via tail vein nick, combined with 47.5 μL of SideStep Lysis & Stabilization Buffer (Agilent Technologies, Santa Clara, CA), and stored at -80˚C until quantitative PCR (qPCR) was performed (see below). 2.5 μL of blood was also added to 1 ml of mBSK medium containing 1X <u>B</u>orrelia <u>a</u>ntibiotic <u>m</u>ixture (BAM) (Monserate, San Diego, CA) to culture bacteria from the blood. On day 14 post-infection, mice were euthanized, and serum was collected for seroconversion analyses (see below). Naïve mice were also sacrificed, and blood was added to lysis/stabilization buffer at a 1:18 ratio for use in qPCR standards (see below). To assess bacterial outgrowth from the blood, blood cultures were analyzed at two-weeks post-collection by dark-field microscopy. 20 fields of view were scanned, and the presence of one or more spirochetes was considered culture positive.

## qPCR for bloodstream bacterial burdens

The method for qPCR to quantify bloodstream bacterial burden was adapted from previous experiments with *B. turicatae* using *flaB* as a target [45,46,63–65]. To generate a qPCR standard, BdWT was cultured to late-exponential growth phase and pelleted by centrifugation at 6,000 x g for 15 min at room temperature. The cells were then resuspended in PBS containing 5 mM MgCl$_2$ (PBS-MgCl$_2$). This wash step was repeated two more times. After the final resuspension in PBS-MgCl$_2$, bacteria were quantified by dark-field microscopy. Tenfold serial dilutions were then made from this resuspension from $10^4$–$10^8$ bacteria/ml in PBS-MgCl$_2$, and dilutions were combined with naïve blood in lysis/stabilization buffer at a 1:19 ratio. For use as a no-template control, nuclease-free water was diluted tenfold in PBS-MgCl$_2$ and added to naïve blood in lysis/stabilization buffer at a 1:19 ratio. In a 96-well real-time PCR reaction plate, 17 μL of master mix consisting of 1X SsoAdvanced Universal Probes Supermix (Bio-Rad Laboratories), 400 nM of each primer (Bd FlaB IDT FWD and Bd FlaB IDT REV) and 300 nM of probe (Bd FlaB IDT PRB) was added to each well. 3 μL of murine blood sample in lysis/stabilization buffer or standard preparation were then added in a 20 μL reaction. qPCR reactions for blood samples and standards were performed in triplicate. The QuantStudio 6 Pro Real-Time PCR System (ThermoFisher Scientific) was used for real-time PCR, and reaction conditions included an initial 2 min, 50˚C step followed by a 10 min, 95˚C hold for polymerase activation. Amplification was then achieved by 40 cycles of DNA denaturation at 95˚C for 15 sec and primer annealing/DNA extension at 60˚C for 60 sec. Following completion of the qPCR reaction, data was imported into GraphPad Prism version 9 (GraphPad Software, San Diego, CA) for analysis. Mean peak bacterial burdens (first peak at ~8 days and second peak at ~12 days) and the <u>a</u>rea <u>u</u>nder the <u>c</u>urve (AUC) for each group of mice were compared using the Kruskal-Wallis test. Difference with *P* values <0.05 were considered statistically significant.

## SDS-PAGE and immunoblotting

SDS-PAGE and immunoblot analyses were performed as previously described [45,66]. Briefly, whole lysates containing the equivalent of 2 x $10^7$ bacteria were separated by SDS-PAGE and transferred to nitrocellulose. Membranes were subsequently probed with rat antiserum recognizing P66 (see above), and horseradish peroxidase (HRP)-conjugated goat anti-rat IgG (Jackson ImmunoResearch Laboratories, West Grove, PA) was used as a secondary antibody. To assess equal loading, membranes were additionally probed for FlaB using chicken anti-*B. burgdorferi* FlaB IgY as the primary antibody and HRP-conjugated donkey anti-chicken IgY (Jackson ImmunoResearch Laboratories) as a secondary antibody [66]. Colorimetric detection of protein was achieved using 4-chloro-1-naphthol as a substrate. To evaluate seroconversion, whole cell lysates were prepared from BdWT bacteria grown to the late exponential growth phase, and volumes containing the equivalent of 2 x $10^7$ cells were separated by SDS-PAGE and transferred to PVDF. Membranes were probed with serum from mice collected at 14 days post-infection (1:1000 dilution), and HRP-conjugated goat anti-mouse IgG (Jackson ImmunoResearch Laboratories) served as the secondary antibody (1:1000 dilution). Reactivity was detected using SuperSignal West Dura extended duration substrate (ThermoFisher Scientific). For seroconversion immunoblots, all images were manually captured for 3 sec without image optimization. Pooled serum from naïve mice and secondary-only exposed controls were also included for the comparison. Precision Plus Protein All Blue Prestained Protein Standard (Bio-Rad Laboratories) was used for the molecular weight standard.

## Results

### Development of a shuttle vector for use in *B. duttonii*

Despite the public health relevance of Old World RF spirochetes, our knowledge regarding the pathogenesis of these bacteria is limited. This is due in part to the lack of techniques to genetically manipulate these pathogens. To begin addressing this gap in the TBRF *Borrelia* field, we sought to develop the requisite genetic tools for use in the Old World RF spirochete, *B. duttonii*. First, we aimed to make a shuttle vector that is competent for replication in both *E. coli* and *B. duttonii*. To identify a putative *B. duttonii* ori for the shuttle vector, the plasmid sequences of strain Ly were scanned for regions homologous to the *B. hermsii* ori used for generation of pBhSV-2 [41]. The cloned origin of replication, derived from *B. duttonii* pl41, included a portion of *bdu_4015*, all of *bdu_4016* and *bdu_4017*, and 137 bp upstream of *bdu_4017*. A kanamycin resistance cassette was generated for *B. duttonii* by fusing the putative promoter for the flagellin gene (*flaB; bdu_150*) to the *aphI* start codon [45,59]. This *aphI* resistance cassette was then ligated together with the MCS from pJD44, an *E. coli* ColE1 ori, and the putative ori from *B. duttonii* pl41, generating the shuttle vector, pBdSV (**Fig 1A**) [41,55,59,60]. An alignment of the ori regions of *B. hermsii* strain DAH lp27 and the pl41 analogous region amplified from *B. duttonii* strain 1120K3 is shown in **S1 Fig** (67.7% identical). A second version of the shuttle vector was additionally made in which the *aphI* cassette was oriented in the opposite direction of that in pBdSV, designated pBdSV$^{CCW}$. To assess if a gene could be successfully expressed from the shuttle vector, a *gfp* reporter was made by fusing the *B. duttonii flaB* promoter to a *gfp* allele and ligating this cassette into the MCS of pBdSV, generating pBdSV::*gfp* [56,57].

To determine if pBdSV, pBdSV$^{CCW}$, and pBdSV::*gfp* were capable of autonomous replication in *B. duttonii*, the three shuttle vectors were transformed into BdWT, creating the strains, BdSV, BdSV$^{CCW}$, and BdSV::*gfp*, respectively. Clones were first confirmed to harbor the shuttle vectors by plasmid recovery in *E. coli*, followed by restriction digest and sequencing of the respective plasmids. PCR was then performed with gDNA from each clone to amplify internal

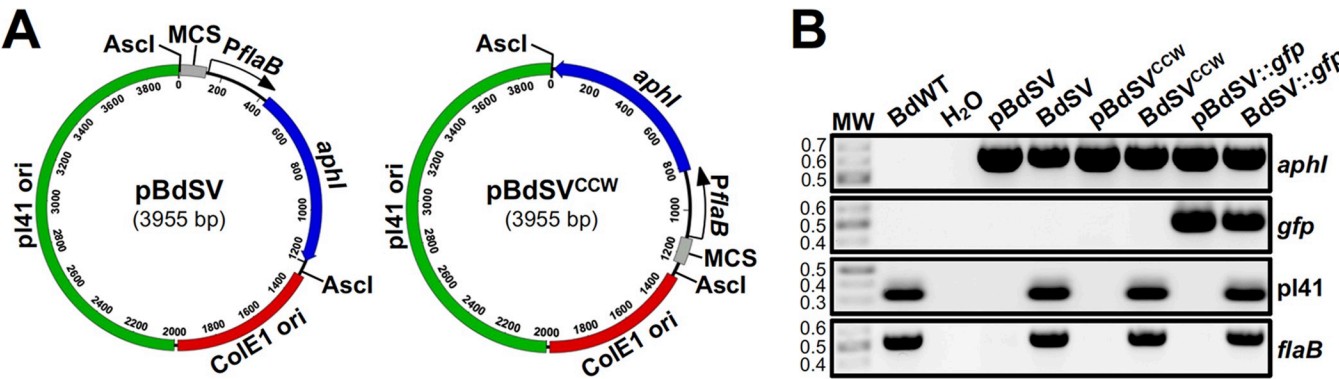

**Fig 1. Generation of the *B. duttonii* shuttle vector.** (A) Diagram illustrating the two *B. duttonii* shuttle plasmids, pBdSV and pBdSV^CCW, developed in this study. In pBdSV^CCW, the MCS and *aphI* resistance cassette from pUAMS363 were ligated in the opposite orientation of this region in pBdSV. (B) Genotypic confirmation of shuttle vector transformants. PCR was performed with BdWT gDNA, the relevant shuttle vectors, and gDNA from *B. duttonii* electroporated with each of the pBdSV constructs to amplify internal regions of *aphI* (624 bp), *gfp* (456 bp), pl41 (326 bp), and *flaB* (531 bp). PCR was also performed with no template (H₂O) as a contamination control. "MW" denotes the DNA standard, and numbers to the left indicate molecular weight in kb.

regions of *aphI*, *gfp*, and *flaB* (**Fig 1B**). As expected, all *B. duttonii* transformants were positive for *aphI*, whereas only BdSV::*gfp* was positive for *gfp*. PCR for *flaB*, which served as an amplification control, resulted in amplicons of the appropriate size in all *B. duttonii* strains. Next, it was possible that transformation of a pl41-derived shuttle vector into *B. duttonii* could result in plasmid incompatibility and cause loss of the endogenous pl41 plasmid [48,59,67–73]. Therefore, a PCR was performed to amplify an internal region of pl41 that was not part of the ori sequence used for generation of pBdSV. All shuttle vector-containing clones screened positive for the endogenous copy of pl41, demonstrating that pBdSV did not displace pl41 in these *B. duttonii* transformants (**Fig 1B**). Finally, to determine that a heterologous gene could be expressed from pBdSV, BdSV::*gfp* was imaged by fluorescence microscopy (**Fig 2**). As expected, BdSV::*gfp* exhibited a bright green fluorescence, but BdWT did not. In all, these results demonstrate successful generation of a shuttle vector capable of autonomous replication in *B. duttonii* that can be used for genetic complementation and ectopic expression experiments (see below).

## Development of a *trans* integration approach for *B. duttonii*

While the intrinsically higher transformation efficiency of shuttle vectors makes them easy to use for *in vitro* expression and complementation experiments, shuttle vectors are not always ideal for *in vivo* applications because the lack of antibiotic selection makes them less stable. Therefore, we sought to develop a *trans* integration approach for use in *B. duttonii* that would recombine gene(s) of interest into the genome [18,74–76], which could prove more optimal for use in animal infection studies. A site on pl165, which is analogous to the large linear plasmids of New World RF *Borrelia*, between the genes *bdu_1105* and *bdu_1106* was chosen as a prospective integration site. This site was selected because it is a large intergenic region (248 bp) between convergently transcribed genes, decreasing the possibility that integration at this site will result in polar mutation effects. To test if a gene could be inserted at this region in pl165, regions flanking the prospective integration site were first amplified, and a segment of pBdSV::*gfp* containing the *aphI* and *gfp* cassettes was excised and ligated between them, creating the integration construct, pBd*gfp* (**Fig 3A**). This construct was then transformed into BdWT, generating Bd*gfp*. Genotypic confirmation of integration in Bd*gfp* was achieved by PCR to amplify internal regions of *gfp* and *aphI*, as well as by a PCR to amplify from upstream of the 5' flanking region used for the integration construct to a site within the *gfp* gene and a

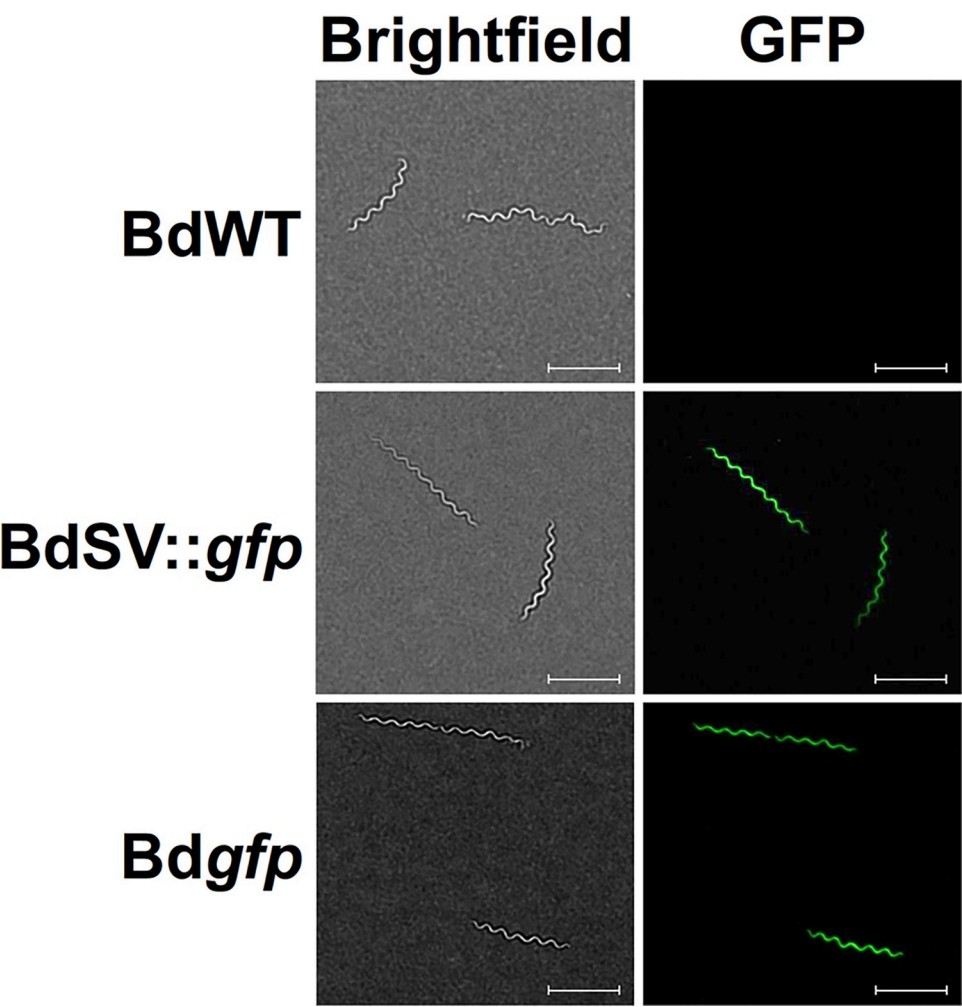

**Fig 2. Fluorescent imaging of *gfp*-expressing *B. duttonii*.** BdWT, BdSV::*gfp*, and Bd*gfp* strains were visualized by either brightfield (left) or fluorescence (right) microscopy. Two biological replicate cultures were examined and shown are representative images from one replicate. Scale bar = 10 μm.

PCR to amplify from within the *aphI* gene to a site downstream of the 3' flanking region used for the integration construct (**Fig 3B**). As expected, Bd*gfp* screened positive for *gfp* and *aphI*, whereas BdWT did not. Additionally, PCR linking sites on pl165 flanking the segments used for the integration construct to sites within the *gfp* and *aphI* ORFs yielded amplicons of the appropriate size in Bd*gfp*, but failed to generate products when BdWT gDNA or pBd*gfp* was used as a template. Finally, a PCR for *flaB*, which served as an amplification control, resulted in products of the appropriate size for BdWT and Bd*gfp*. Fluorescence microscopy was then used to demonstrate expression of *gfp* in Bd*gfp* (**Fig 2**). As expected, Bd*gfp* fluoresced green, whereas BdWT exhibited no fluorescence. In all, these results confirm that the integration site selected within pl165 can be used for insertion of genes, validating use of this approach for genetic expression and complementation experiments.

### Allelic exchange mutagenesis in *B. duttonii*

The above data confirmed successful transformation of *B. duttonii* with the shuttle vector and pl165-targeting integration construct. As future work will aim to characterize bacterial factors

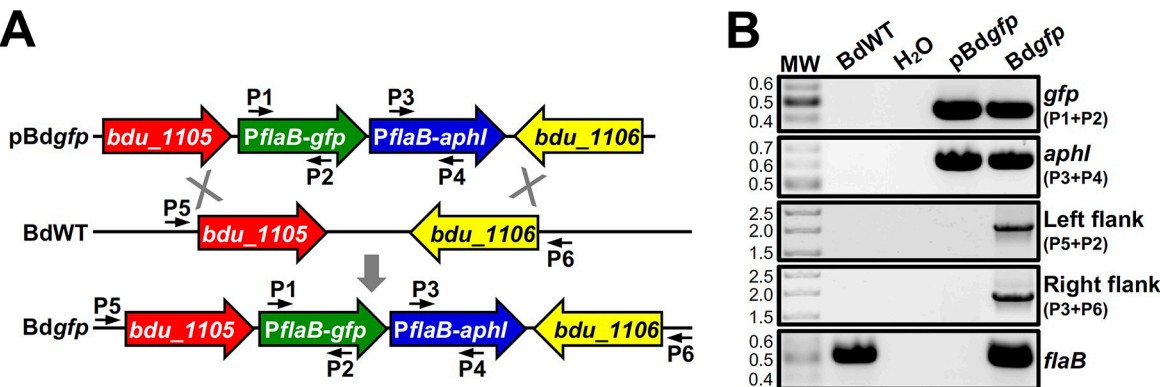

**Fig 3. Use of the *trans* integration approach to generate *gfp*-expressing *B. duttonii*.** (A) Generation of Bd*gfp*. BdWT was transformed with an allelic exchange construct (pBd*gfp*) to insert P*flaB-gfp* and P*flaB-aphI* cassettes into pl165, generating Bd*gfp*. Primers used for PCR in Fig 3B are denoted with numbered smaller arrows. (B) Genotypic confirmation of Bd*gfp*. PCR was performed with BdWT gDNA, pBd*gfp* plasmid, and Bd*gfp* gDNA to amplify an internal segment of *gfp* (P1+P2; 456 bp), an internal segment of *aphI* (P3+P4; 624 bp), a segment spanning regions of *bdu_1105* and P*flaB-gfp* (P5+P2; 1967 bp), a segment spanning regions of P*flaB-aphI* and *bdu_1106* (P3+P6; 1937 bp), and an internal segment of *flaB* (531 bp). PCR was also performed with no template (H$_2$O) as a contamination control. "MW" denotes DNA standard, and numbers to the left indicate molecular weight in kb.

required for the *B. duttonii* enzootic cycle, it was also necessary to develop and confirm methods for allelic exchange mutagenesis in *B. duttonii*. As a proof of principle, we chose to mutate a homolog of a gene that is known to be required for mammalian infection by the related LD spirochete, *B. burgdorferi*. Specifically, P66 can function as both an adhesin and a porin in *B. burgdorferi*, so the gene encoding the *B. duttonii* P66 homolog (BDU_604) was targeted for allelic exchange mutagenesis [51,52,77–85]. BDU_604 and P66 of *B. burgdorferi* strain B31 are 53.7% identical and their genes are colinear in the chromosome. A gentamicin resistance marker was first adapted for use in *B. duttonii* by amplifying the putative promoter region for the flagellar basal body rod (*flgB*) gene and fusing it to the start codon of the *aacC1* ORF [41,46,47]. The *B. duttonii*-adapted *aacC1* marker was subsequently ligated between regions flanking *bdu_604*, replacing a 1750-bp region of the ORF and generating the mutational construct, pBdΔ*p66*. This construct was then electroporated into BdWT, and transformants were selected with gentamicin (**Fig 4A**).

A *B. duttonii p66* mutant clone, designated BdΔ*p66*, was confirmed by PCR with primer pairs that amplified a region within the deleted portion of the *p66* ORF or an internal region of *aacC1* (**Fig 4B**). An internal region of *flaB* was also amplified as a positive control. As expected, PCR for the internal region of *bdu_604* resulted in an amplicon of the appropriate size for BdWT, while no amplicon was observed for BdΔ*p66*. Alternatively, BdΔ*p66* screened positive for the *aacC1* marker, while BdWT did not. Additionally, PCR for a region flanking the site of mutation resulted in products of the appropriate size for BdWT and BdΔ*p66*. Finally, PCR for an internal region of *flaB* resulted in amplicons for both BdWT and BdΔ*p66* strains (**Fig 4B**).

Genetic complementation was used to restore expression of *p66* in BdΔ*p66*. There are three commonly used methods for genetic complementation; i) complementation in *cis* by replacing the mutated gene with a wild-type allele, ii) complementation in *trans* by transforming a plasmid that harbors a wild-type copy of the gene into the mutant strain, and iii) complementation in *trans* by inserting a wild-type copy of the gene into a stable genomic element (e.g., endogenous plasmid or chromosome) in the mutant strain [19]. We encountered complications cloning the *p66* region into the pl165-targeting *trans* integration construct (e.g., pUAMS376), so *cis* complementation and shuttle-vector-based *trans* complementation were pursued in this study. To generate the *p66* shuttle vector complementation construct, the *p66* ORF and 540-bp upstream of the coding region were amplified from BdWT and cloned into pBdSV to generate

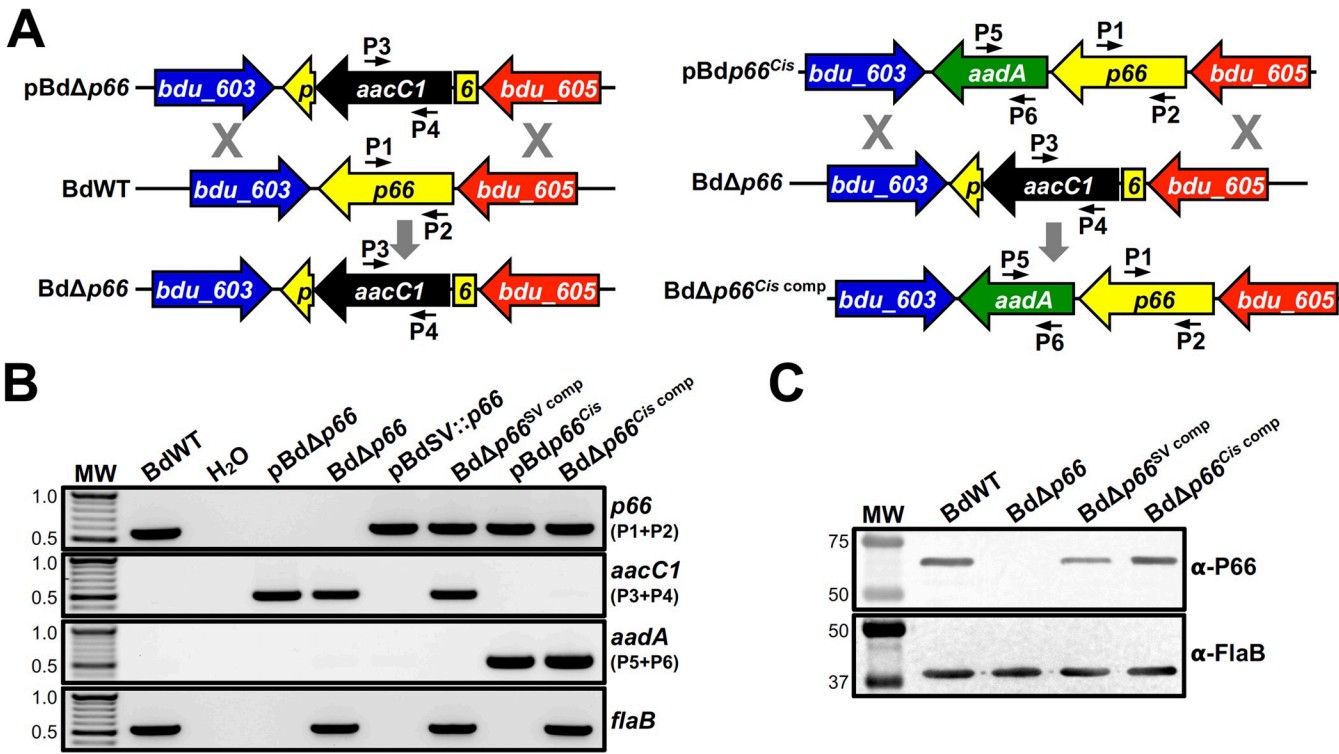

**Fig 4. Mutation and complementation of *B. duttonii p66*.** (A) Schematic representation of the relevant regions of the chromosomes and relevant constructs used to generate BdΔ*p66* mutant and BdΔ*p66*^Cis comp^. BdWT was transformed with an allelic exchange construct, pBdΔ*p66*, to replace an internal region of *p66* with P*flgB-aacC1* marker, generating BdΔ*p66*. To restore *p66* in BdΔ*p66*, the mutant was transformed with pBd*p66*^Cis^ to replace the P*flgB-aacC1* with an intact copy of *p66* and P*flaB-aadA*, generating BdΔ*p66*^Cis comp^. Numbered arrows represent approximate locations of primers used in panel B. (B) Genotypic confirmation of BdΔ*p66*, BdΔ*p66*^SV comp^, and BdΔ*p66*^Cis comp^. PCR was performed with gDNAs isolated from BdWT, BdΔ*p66*, BdΔ*p66*^SV comp^, and BdΔ*p66*^Cis comp^. Plasmids pBdΔ*p66*, pBdSV::*p66*, and pBd*p66*^Cis^ were included as positive amplification controls, and PCR was also performed with no template (H₂O) as a purity control. Diagnostic PCR, identified on right, was designed to amplify internal regions of *p66* (P1+P2; 565 bp), *aacC1* (P3+P4; 489 bp), *aadA* (P5+P6; 463 bp), or *flaB* (531 bp). "MW" denotes the DNA standard, and numbers to the left indicate molecular weight in kb. (C) Immunoblot confirmation of BdΔ*p66*, BdΔ*p66*^SV comp^, and BdΔ*p66*^Cis comp^. Whole-cell lysates, prepared from bacteria in late-exponential phase cultures, were separated by SDS-PAGE and transferred to a nitrocellulose membrane. Membranes were then probed with antiserum or antibody against P66 or FlaB, respectively. Antiserum/antibodies used to detect the respective proteins are indicated to the right. "MW" denotes the protein standard, and numbers to the left indicate molecular weight in kDa.

pBdSV::*p66*. For the *cis* integration *p66* complementation construct, an *aadA* resistance marker was first adapted for *B. duttonii* by fusing the promoter region of BdWT *flaB* to the *aadA* ORF to generate P*flaB-aadA* [41,46,61,86]. The genomic regions flanking *bdu_604*/*p66*, corresponding to a portion of *bdu_602* with all of *bdu_603* and all of *bdu_604* with a portion of *bdu_605* were then amplified, and the P*flaB-aadA* marker was then ligated into a unique AscI site introduced downstream of the *p66* ORF. The resulting *p66 cis* complementation construct was designated pBd*p66*^Cis^ (**Fig 4A**). pBdSV::*p66* and pBd*p66*^Cis^ were electroporated into BdΔ*p66*, and transformants were designated BdΔ*p66*^SV comp^ and BdΔ*p66*^Cis comp^, respectively. BdΔ*p66*^SV comp^ and BdΔ*p66*^Cis comp^ were initially confirmed by PCR, and *p66* was detected in both complemented strains (**Fig 4B**). In BdΔ*p66*^Cis comp^, the *aadA* marker was detected instead of *aacC1*, whereas *aacC1* was detected in BdΔ*p66*^SV comp^. The pBdSV::*p66* shuttle vector was also recovered from BdΔ*p66*^SV comp^ and confirmed by restriction digestion and sequencing. Finally, loss of P66 production in BdΔ*p66* and restoration of P66 in BdΔ*p66*^SV comp^ and BdΔ*p66*^Cis comp^ was confirmed by immunoblot (**Fig 4C**). In all, these results indicate the first successful mutagenesis of a gene in an Old World RF spirochete and confirm that allelic exchange is a viable option for targeted mutagenesis in *B. duttonii*.

## The BdΔ*p66* mutant is attenuated in a murine model of RF

Murine infection models are commonly used to determine if RF spirochete mutants are competent for mammalian infection, [41–50,65]. We therefore adapted the previously described murine infection model for the New World RF *Borrelia*, *B. turicatae*, for use with *B. duttonii* [44–47,50]. Groups of four mice were intradermally/subcutaneously inoculated with $10^2$ spirochetes of BdWT, BdΔ*p66*, BdΔ*p66*$^{SV comp}$, and BdΔ*p66*$^{Cis comp}$. Blood samples were collected on days 3–14 and analyzed by qPCR to quantify the bacterial number in the bloodstream. Mice infected with BdWT experienced two recurring bouts of high bacteremia with peaks of ~$10^8$ bacteria/ml of blood (**Fig 5A**). In contrast, two mice infected with BdΔ*p66* failed to reach detectable levels of bacteremia on any day post-infection. The other two BdΔ*p66*-infected mice showed two bacteremia peaks, but the bacteria reached levels lower than BdWT (**Fig 5B**). In these latter two mice, the first bacteremic peaks reached $10^5$–$10^6$ bacteria/ml and ~$10^6$ bacteria/ml were detected during the second peak. Statistical analyses comparing the mean peak bacterial burdens or AUC values between BdWT- and BdΔ*p66*-infected mice confirmed statistically significant differences (**S2 Fig**). To fulfill molecular Koch's postulates and confirm the requirement for P66 during *B. duttonii* mammalian infection, genetic complementation was used to restore expression of *p66* in BdΔ*p66* and determine if the infection defect could be reversed. For the genetically complemented strains, bacterial burdens were comparable to BdWT in three of the four mice infected with BdΔ*p66*$^{SV comp}$ (**Fig 5C**) and in all four mice infected with BdΔ*p66*$^{Cis comp}$ (**Fig 5D**). Statistical comparison of the data from the BdΔ*p66*$^{Cis comp}$- and BdΔ*p66*-infected mice confirmed significant differences only in the mean peak bacterial burdens. Neither the AUC values or peak burdens between the BdΔ*p66*$^{SV comp}$- and BdΔ*p66*-infected groups were statistically significant (**S2 Fig**). Daily blood samples were also collected and inoculated into mBSK medium to culture *B. duttonii* from the bloodstream. This approach is predicted to be more sensitive with an estimated limit of infection (LOD) of ~4 x $10^2$ live bacteria/ml [45,46]. BdWT was cultured from the blood of infected mice on multiple days post-infection, but BdΔ*p66* spirochetes could only be cultured from the two mice that showed the bacteremia peaks. Furthermore, all the mice infected with BdΔ*p66*$^{SV comp}$ and BdΔ*p66*$^{Cis comp}$ were culture positive on at least one day during infection. Seroconversion results agreed with the qPCR results (**S3 Fig**). Specifically, the two BdΔ*p66*-infected and one BdΔ*p66*$^{SV comp}$-infected mice that didn't show detectable levels of bacteremia after infection also did not produce antibodies that reacted against BdWT lysates. Taken together, these results support the notion that P66 is required for optimal mammalian infection and demonstrate our ability to adapt our murine model of RF used for the New World RF bacterium, *B. turicatae*, to study pathogenesis of the Old World RF spirochete, *B. duttonii*.

## Discussion

Due to the phylogenetic divergence of Old World and New World RF spirochetes, studying representatives from each of these distinct groups of *Borrelia* could reveal differences in factors required during their respective enzootic cycles [16,26,31,32,34,87,88]. However, to date, genetic manipulation of RF spirochetes has only been described in New World RF *Borrelia* [41–50,65]. To address this gap in the literature, we sought to develop the requisite genetic tools and techniques to study the Old World RF bacterium, *B. duttonii*. Allelic exchange mutagenesis is the most common method for mutation in *Borrelia* spp. and was therefore used to attempt inactivation of a *B. duttonii* gene [18,19,41,47,74,75]. To test this approach, the gene encoding the homolog of the known *B. burgdorferi* virulence factor, P66, was targeted for inactivation. P66 in LD *Borrelia* functions as both a porin and an integrin binding adhesin that facilitates transendothelial migration and promotes *B. burgdorferi* dissemination [51,52,77–

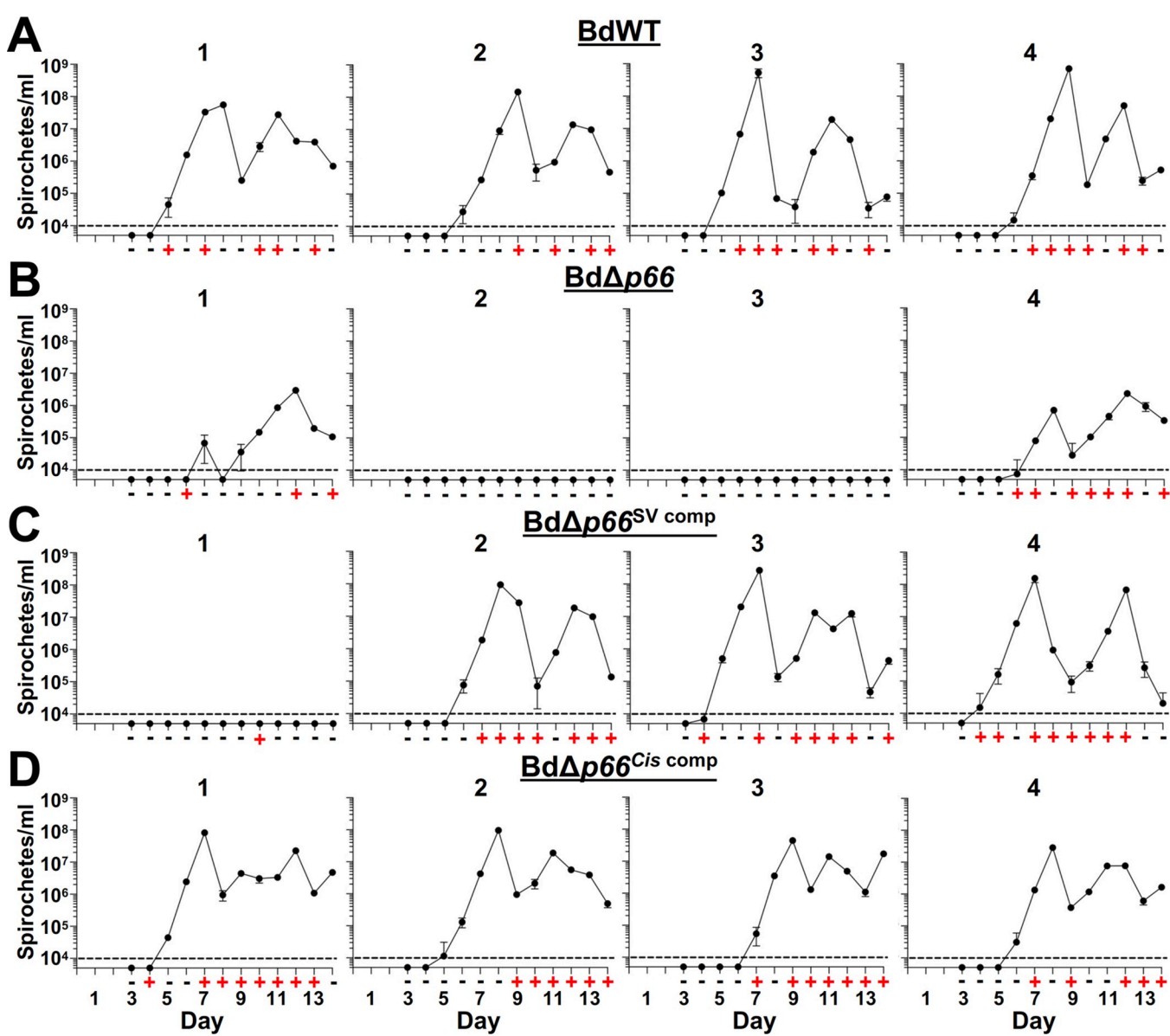

**Fig 5. Murine infection phenotype of the *B. duttonii p66* mutant and complemented clones.** Groups of four mice were intradermally inoculated with $10^2$ (A) BdWT, (B) BdΔ*p66*, (C) BdΔ*p66*[SV comp], and (D) BdΔ*p66*[Cis comp] spirochetes. On days 3–14 post-infection, bacterial levels in the bloodstream of each mouse were quantified by qPCR. The dashed line indicates the LOD for this assay ($10^4$ spirochetes/ml). Daily blood samples were also collected and inoculated into mBSK to culture *B. duttonii* from the bloodstream. Cultures were analyzed up to two-weeks post-collection by dark-field microscopy and results for each day are indicated below the X-axis; +, positive for spirochetes and -, no spirochetes observed. Numbers above the graphs indicate individual mice in each experimental group, and error bars represent standard error of the mean (SEM).

85], thus we hypothesized that a *B. duttonii p66* mutant would not be fully infectious. PCR and immunoblot analyses confirmed successful mutation of *p66*, and the BdΔ*p66* mutant was found to be attenuated in a murine model of RF. Importantly, genetic complementation was used to restore P66 production by BdΔ*p66*, which also restored normal infectivity to the mutant and confirmed P66 contributes to mammalian infection by *B. duttonii*. This manuscript is the first to establish an essential role for a P66 homolog during mammalian infection by any RF spirochete. Little is known about the *in vivo* function(s) of P66 in RF *Borrelia*, but Bárcena-Uribarri et al. demonstrated that P66 homologs in three LD spirochetes

(*B. burgdorferi*, *Borrelia afzelii*, and *Borrelia garinii*) and three RF species [*B. duttonii* (Old World), *B. recurrentis* (Old World), and B. hermsii (New World)] have highly conserved amino acid domains and similar porin biophysical properties *in vitro*, suggesting that P66 homologs play similar roles across *Borrelia* species [83]. Future studies will be required to assess the possible function of RF *Borrelia* P66 homologs in integrin binding and to assess whether potential roles in integrin binding and channel formation are essential during the enzootic cycle.

Two methods were used to genetically complement the BdΔ*p66* mutant in this study. For the first method, a *B. duttonii* shuttle vector was generated and *p66* along with its putative pro-moter region was inserted into the engineered MCS. While several shuttle vectors have been made for use in the LD spirochetes, few exist for use in RF spirochetes [18,19,41,45,47,74–76,89,90]. In fact, prior to this study, shuttle vectors had only been generated for the New World RF spirochetes, *B. hermsii* and *B. turicatae* [41,45]. Borrelial shuttle vectors are generally useful for *in vitro* experiments, but these plasmids have varying stability *in vivo* in the absence of antibiotic selection. In fact, genetic complementation of BdΔ*p66* using pBdSV::*p66* resulted in one mouse that did not seroconvert or produce measurable bacteremia in the blood, despite having a positive blood culture on day 10, potentially suggesting loss of the shuttle vector from the population infecting this mouse. As expected, genetic complementation of BdΔ*p66* using a *cis* integration approach resulted in full restoration of mammalian infection in all infected mice, validating the use of this approach for future *in vivo* complementation experiments using our murine model of RF.

In addition to being valuable for use in genetic complementation, the shuttle vector and *trans* integration methods described herein have potential for other future uses, including het-erologous expression experiments. This is evidenced by our ability to express a *gfp* allele using both approaches *in vitro*. Expression of *gfp* has also been used in other *Borrelia* to study patho-gen-host interactions, and *Borrelia*-tick interactions have been visualized using both *B. burg-dorferi* and *B. turicatae* strains expressing *gfp* [50,91–93]. Recently, we have developed an *Ornithodoros moubata*-mouse model of transmission/infection to study tick acquisition, colo-nization, and transmission [94]. Use of the *gfp*-expressing Bd*gfp* reported herein should allow visualization of *B. duttonii*-tick interactions. Given the evolutionary divergence of Old World and New World RF spirochetes, these studies could reveal differences in the vector phase of the enzootic cycle. While use of *gfp*-expressing strains to study *B. duttonii*-host interactions should provide valuable information, this represents just one of many possible applications of our shuttle vector and integration strategies for introduction and ectopic expression of genes in *B. duttonii* [18,19].

This study lays the foundation for the application of genetic tools to study Old World RF spirochetes, but more work is required to advance the available armamentarium of genetic tools. We successfully adapted three commonly used antibiotic resistance markers in *Borrelia* for use in *B. duttonii*; the *Pseudomonas aeruginosa* Tn*1696*-derived *aacC1* gentamicin marker, the *E. coli* Tn*903*-derived *aphI* kanamycin marker, and the *Shigella flexneri* plasmid R100-derived *aadA* streptomycin resistance marker, [41,86,95,96] Future work may require the development of additional resistance markers. Therefore, we plan to expand this available repertoire of *B. duttonii* markers by adapting commonly used resistance markers in other *Bor-relia* systems, including the *Aspergillus terreus*-derived blasticidin resistance marker, *bsd*, and the *E. coli*-derived hygromycin marker, *hph* [97,98]. Additionally, while we proposed the potential use of *gfp* as a reporter gene to visualize *B. duttonii* in a given environment (e.g., tick tissues), other reporter genes have been adapted for use in *Borrelia* spp. for similar purposes, as well as to monitor gene expression or track cellular localization of various proteins [18,19,92]. These reporters include chloramphenicol acetyl transferase (CAT), β-galactosidase

(LacZ), firefly luciferase, yellow fluorescent protein (YFP), cyan fluorescent protein (CFP), red fluorescent protein (RFP), and orange fluorescent protein (dTomato) [89,91,93,98–105]. Future work will also involve adaptation of the genes encoding these other reporters for use in *B. duttonii*. Finally, development of an inducible expression system would provide the ability to overexpress a gene and determine phenotypes associated with altered expression *in vitro* or *in vivo*. We developed one of the first inducible expression systems available for use in *B. burgdorferi* and have recently adapted this system for use in the New World RF spirochete, *B. turicatae* [45,99]. Given our prior success, we envision successful adaptation of this system for use in *B. duttonii* as well.

One significant obstacle to genetic manipulation of *Borrelia* spp. is the low transformation efficiency relative to other bacterial systems [18,19,41,47,74,75]. With *B. duttonii*, we have also observed low transformation frequencies, with ~20% of transformations being successful. Therefore, future work will aim at increasing transformation efficiency. Possible changes to test include altering bacterial density/growth phase at the time of harvest, changing the amounts and types of washes performed during electrocompetent cell preparation, and linearization of suicide constructs prior to electroporation. One possible reason for low transformation efficiency observed in *B. duttonii* though could be the presence of restriction modification systems, which is observed in *B. burgdorferi* [106–109]. Interestingly, mutation of at least one New World RF spirochete restriction modification gene does not impact infectivity [49]. Therefore, it may be possible to generate a strain in which all restriction modification systems have been inactivated that is fully infectious. This strain could then serve as the "wild-type" parent to generate mutants to assess the requirement for certain factors throughout the enzootic cycle. The development of additional resistance markers would be critical to use this strategy however (see above). As an alternative, *in vitro* methylation could be performed on constructs to transform into *B. duttonii*, as has been done to increase transformation efficiency in *B. burgdorferi* [108]. Finally, alternative means of transformation could be explored to assess differences in efficiency relative to electroporation. To date, only one other means of transformation has been explored in a *Borrelia* species, transformation and storage solution (TSS) transformation [110,111]. While this procedure is not commonly used, it has worked successfully in *B. burgdorferi*, and, therefore, may serve as a viable alternative to electroporation. Changes in transformation protocols, methods to circumvent *B. duttonii* restriction-modification systems, and alternative means of transformation will all be the subject of future work.

Herein, we developed the first genetic tools and techniques to begin studying factors required for the enzootic cycle of an Old World RF spirochete using the bacterium, *B. duttonii*. In all, this study lays the groundwork to begin expanding our knowledge regarding the enzootic cycle and pathogenesis of Old World RF spirochetes. An increased understanding of these neglected, public health-relevant pathogens could ultimately provide better options for prevention and treatment of RF in developing countries where these spirochetes are among the leading causes of bacterial illnesses.

## Supporting information

**S1 Table. Primers used in this study.**
(PDF)

**S1 Fig. Alignment of *B. hermsii* lp27 and the analogous pl41 ori region amplified from *B. duttonii*.** Shaded boxes indicate nucleotide identity. "Bh" and "Bd" denote *B. hermsii* strain DAH and *B. duttonii* strain 1120K3 sequences, respectively.
(PDF)

**S2 Fig. Statistical analysis of murine infection data.** Mean peak bacterial burdens (first peak at ~8 days and second peak at ~12 days) and the area under the curve (AUC) for each group of mice were compared using the Kruskal-Wallis test. Values are graphed as the geometric means with geometric standard deviations. ns, not significant; *, $P < 0.05$; and **, $P < 0.005$. (PDF)

**S3 Fig. Seroconversion in mice infected with *B. duttonii p66* mutant and complemented clones.** Whole cell lysates of BdWT bacteria were separated by SDS-PAGE, transferred to PVDF membranes, and probed with serum from mice collected at 14 days post-infection. Images were manually captured for the same length of time. A pooled naïve mouse sample and secondary-only exposed controls were also included in the analysis; e.g., "Naïve" and "2˚ only", respectively. "MW" denotes the protein standard, and numbers to the left indicate molecular weight in kDa. (PDF)

## Acknowledgments

The authors thank Jennifer Johnson in the UAMS DNA and Next-Generation Sequencing Core facility for technical assistance. We also acknowledge Chris Johnsten for his invaluable expertise and guidance throughout this project.

## Author Contributions

**Conceptualization:** Clay D. Jackson-Litteken, Jon S. Blevins.

**Data curation:** Clay D. Jackson-Litteken, Wanfeng Guo, Brandon A. Hogland, C. Tyler Ratliff, Marissa S. Fullerton, Daniel E. Voth, Jon S. Blevins.

**Formal analysis:** Clay D. Jackson-Litteken, Wanfeng Guo, Brandon A. Hogland, C. Tyler Ratliff, LeAnn McFadden, Jon S. Blevins.

**Funding acquisition:** Jon S. Blevins.

**Investigation:** Clay D. Jackson-Litteken, Jon S. Blevins.

**Methodology:** Clay D. Jackson-Litteken, Jon S. Blevins.

**Project administration:** Jon S. Blevins.

**Resources:** Jon S. Blevins.

**Supervision:** Jon S. Blevins.

**Validation:** Clay D. Jackson-Litteken, Wanfeng Guo, Brandon A. Hogland, C. Tyler Ratliff, LeAnn McFadden, Marissa S. Fullerton, Daniel E. Voth, Ryan O. M. Rego, Jon S. Blevins.

**Visualization:** Clay D. Jackson-Litteken, Wanfeng Guo, Brandon A. Hogland, C. Tyler Ratliff, LeAnn McFadden, Marissa S. Fullerton, Daniel E. Voth, Ryan O. M. Rego, Jon S. Blevins.

**Writing – original draft:** Clay D. Jackson-Litteken, Wanfeng Guo, Brandon A. Hogland, C. Tyler Ratliff, Jon S. Blevins.

**Writing – review & editing:** Clay D. Jackson-Litteken, Wanfeng Guo, Brandon A. Hogland, C. Tyler Ratliff, LeAnn McFadden, Marissa S. Fullerton, Daniel E. Voth, Ryan O. M. Rego, Jon S. Blevins.

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
