## [Decision Letter · Decision Letter 0]

24 Jun 2024

Dear Dr. Blevins,

Thank you very much for submitting your manuscript "Development and validation of systems for genetic manipulation of the Old World tick-borne relapsing fever spirochete, Borrelia duttonii" for consideration at PLOS Neglected Tropical Diseases. As with all papers reviewed by the journal, your manuscript was reviewed by members of the editorial board and by several independent reviewers. The reviewers appreciated the attention to an important topic. Based on the reviews, we are likely to accept this manuscript for publication, providing that you modify the manuscript according to the review recommendations. 

Sincerely,

Travis J Bourret

Academic Editor

Georgios Pappas

Section Editor

Reviewer's Responses to Questions

**Key Review Criteria Required for Acceptance?**

**Methods**

-Are the objectives of the study clearly articulated with a clear testable hypothesis stated?

-Is the study design appropriate to address the stated objectives?

-Is the population clearly described and appropriate for the hypothesis being tested?

-Is the sample size sufficient to ensure adequate power to address the hypothesis being tested?

-Were correct statistical analysis used to support conclusions?

-Are there concerns about ethical or regulatory requirements being met?

Reviewer #1: (No Response)

Reviewer #2: The methods are straightforward and well described.

1. Despite showing the SEM, the animal infection study shown in Fig. 5 requires statistical analysis.

2. (lines 310-311) Mention that flaB is the PCR target.

**Results**

-Does the analysis presented match the analysis plan?

-Are the results clearly and completely presented?

-Are the figures (Tables, Images) of sufficient quality for clarity?

Reviewer #1: (No Response)

Reviewer #2: The results are clearly presented.

3. How similar are the origins of replication used for the shuttle vectors from B. hermsii and B. duttonii? Perhaps a supplemental figure with a sequence alignment would be appropriate.

3. The figure legends for Figs. 2 and 3 are swapped.

5. (line 410) Is pl165 homologous, or analogous, to lp200 in the New World Borrelia?

6. Did mouse 1 of the SV complement lose the plasmid? There is one positive culture (which theoretically could be assayed).

**Conclusions**

-Are the conclusions supported by the data presented?

-Are the limitations of analysis clearly described?

-Do the authors discuss how these data can be helpful to advance our understanding of the topic under study?

-Is public health relevance addressed?

Reviewer #1: (No Response)

Reviewer #2: The conclusions are valid and clearly presented, as is the significance of Old World RF.

7. (line 603) Erythromycin is used to treat TBRF in children, so introducing erythromcyin resistance might not be prudent; in addition, ermC is not a great marker and has only seldomly been using in LD spirochetes (selection is challenging because the bacteria are susceptible to the antibiotic and ermC does not confer high-level resistance).

**Editorial and Data Presentation Modifications?**

Reviewer #1: (No Response)

Reviewer #2: Note that PLOS does not permit “data not shown.” You could put these in Supplemental Data.

(throughout manuscript) “Borrelia spirochetes” seems awkward: “RF spirochetes”, “LD spirochetes” and “Borrelia spp.” would be more appropriate.

(line 38) “Borrelia burgdorferi p66 gene, which encodes a virulence factor,” instead of “Borrelia burgdorferi virulence factor, p66,”

(line 140) Perhaps a bit of trivia: “LB” is an abbreviation for “Lysogeny Broth” (see Bertani, 2004, J. Bacteriol. 186:595-600).

(throughout Methods) Restriction enzyme names are not italicized.

(throughout manuscript) “Confirmatory PCRs” is awkward; perhaps “Mutations or complementing DNA was confirmed by PCR and products” (line 238) and just “PCR” (lines 243 and 259). Also just “PCR” instead of “PCRs” and “a PCR”.

(lines 315 and 318) “Tenfold” instead of “10-fold”.

(lines 380-381) delete “Borrelia spirochetes maintain very low copy numbers of each plasmid in the genome,”.

(line 547) “However” instead of “owever”.

(line 572) “bacteria” instead of “bacterium” (or better yet “spirochetes”).

Reference 100 should be Alverson et al., 2003 (https://doi.org/10.1046/j.1365-2958.2003.03537.x) not Alverson and Samuels, 2002.

**Summary and General Comments**

Reviewer #1: (No Response)

Reviewer #2: The development of a genetic manipulation system for the Old World RF spirochetes is an important contribution; this study has been carefully done and is clearly presented. Besides for the statistical analysis of the animal experiment, I have only minor and editorial comments.

The rationale for mutating p66 is to serve as a proof-of-concept demonstrating the methodology, but this is also intrinsically an important genetic study: P66, an interesting bifunctional outer membrane porin/intergrin-binding adhesin, has not been previously experimentally evaluated in any RF spirochete. This work strongly suggests that P66 has a crucial role in RF pathogenesis, albeit mechanistic details are absent.

PLOS authors have the option to publish the peer review history of their article (what does this mean?). If published, this will include your full peer review and any attached files.

Reviewer #1: No

Reviewer #2: No

Figure Files:

Data Requirements:

Reproducibility:

References

---

## [Editor Report · Decision Letter 1]

8 Jul 2024

Dear Dr. Blevins,

We are pleased to inform you that your manuscript 'Development and validation of systems for genetic manipulation of the Old World tick-borne relapsing fever spirochete, Borrelia duttonii' has been provisionally accepted for publication in PLOS Neglected Tropical Diseases.

Best regards,

Travis J Bourret

Academic Editor

Georgios Pappas

Section Editor

---

## [Editor Report · Acceptance letter]

17 Jul 2024

Dear Dr. Blevins,

We are delighted to inform you that your manuscript, "Development and validation of systems for genetic manipulation of the Old World tick-borne relapsing fever spirochete, Borrelia duttonii," has been formally accepted for publication in PLOS Neglected Tropical Diseases.

Best regards,

Shaden Kamhawi

co-Editor-in-Chief

Paul Brindley

co-Editor-in-Chief
